# Replacement Instead of Discontinuation of Bacillus Calmette-Guérin Instillation in Non-Muscle-Invasive Bladder Cancer

**DOI:** 10.3390/cancers15041345

**Published:** 2023-02-20

**Authors:** Po-Ting Lin, Wei-Kang Hung, Ying-Hsu Chang, Ming-Li Hsieh, Chung-Yi Liu, Liang-Kang Huang, Yuan-Cheng Chu, Hung-Cheng Kan, Po-Hung Lin, Kai-Jie Yu, Cheng-Keng Chuang, Chun-Te Wu, See-Tong Pang, I-Hung Shao

**Affiliations:** 1Division of Urology, Department of Surgery, Chang Gung Memorial Hospital, Linkou Branch, Taoyuan 333, Taiwan; 2Department of Medicine, Chang Gung University, Taoyuan 333, Taiwan; 3Department of Urology, New Taipei Municipal TuCheng Hospital, Chang Gung Memorial Hospital and Chang Gung University, New Taipei 236, Taiwan; 4Cancer Genome Research Center, Chang Gung Memorial Hospital, Linkou Branch, Taoyuan 333, Taiwan

**Keywords:** bladder cancer, intravesical instillation, NMIBC, BCG, bladder recurrence

## Abstract

**Simple Summary:**

Most patients with intermediate- and high-risk non-muscle-invasive bladder cancer (NMIBC) fail to complete bacillus Calmette-Guérin (BCG) instillation due to toxicity or shortage. Our retrospective study aimed to assess the efficacy of intravesical chemotherapy replacement at the time of BCG interruption. We confirmed the chemotherapy replacement group had a better three-year bladder recurrence-free survival (RFS) than the BCG discontinuation group. Chemotherapy replacement is then a helpful strategy in patients with NMIBC facing BCG treatment stoppage.

**Abstract:**

Background: To evaluate the efficacy of intravesical chemotherapy replacement in patients with intermediate- and high-risk non-muscle-invasive bladder cancer (NMIBC), who underwent bacillus Calmette-Guérin (BCG) instillation but discontinued due to global shortages or toxicity of BCG. Methods: This retrospective study included patients with intermediate- and high-risk NMIBC who received BCG intravesical instillation. Those who discontinued the treatment were divided into the pure BCG group and chemotherapy replacement group. Comparisons between these groups were performed. The primary endpoint was bladder recurrence-free survival (RFS). Results: A total of 480 patients were included. Baseline characteristics were similar between groups, but the total instillation times were higher in the chemotherapy replacement group than in the pure BCG group (n = 14.9 vs. 10.5). The chemotherapy replacement group had a better three-year RFS (*p* = 0.022). On multivariate analysis, the pure BCG group had significantly increased all-time and 3-year recurrences (hazard ratio 2.015 and 2.148) compared to the chemotherapy replacement group. Conclusions: Chemotherapy replacement has a better three-year RFS than no instillation in patients with intermediate- and high-risk NMIBC who received BCG instillation but facing treatment stoppage.

## 1. Introduction

Bladder cancer accounts for approximately 573,278 new cases diagnosed per year and is the tenth most common cancer worldwide. The age-standardized incidence (per 100,000 person/years) is 5.6 considering both genders (men, 9.5; women, 2.4) [1].

The tumor stage of bladder cancer is assigned by the depth of the bladder wall invasion. Tumors with muscle layer invasion or beyond are considered muscle-invasive bladder cancer (MIBC). It could be further divided into non-metastatic or metastatic disease, accounting for approximately 4% of the newly diagnosed bladder cancers. In contrast, the cancers confined to the urothelium and the lamina propria layer are called non-muscle-invasive bladder cancer (NMIBC), accounting for approximately 75% of newly diagnosed bladder cancer [2,3].

In treating NMIBC, transurethral resection of bladder tumor (TUR-BT) is the first step that improves clinical and pathological features such as tumor size, number, stage, and histology grade [3]. Based on these variables, patients can be stratified into low, intermediate, and high risk. The prognosis of each risk group is different, thus requiring different treatment recommendations [4]. The European Association of Urology (EAU) guidelines suggest only a single installation of intravesical chemotherapy in low-risk groups, while further induction and maintenance of adjuvant intravesical agents for intermediate and high-risk groups [5]. Among the adjuvant treatments, intravesical bacillus Calmette-Guérin (BCG) immunotherapy is considered superior to the other agents in preventing recurrence and progression [5]. However, only 16% of patients tolerate the whole BCG treatment course due to significant toxicity [6].

Besides the toxicity of BCG that hinders the continuity of BCG, another added problem is BCG shortage. In the past decade, several events, such as BCG plant closures and manufacturer withdrawal have reduced BCG production and led to a supply-demand mismatch [7]. Strategies including a change of BCG schedule or adopting intravesical chemotherapy are discussed to cope with the BCG instillation discontinuation.

In this retrospective study, we compared the outcome of patients with NMIBC who received BCG instillation after suffering from discontinuation due to side effects or shortage, and those who received initial BCG instillation and discontinued then shifted to intravesical chemotherapy. We aimed to investigate whether intravesical chemotherapy replacement is helpful in patients who discontinued BCG either due to intolerant side effects or BCG shortage.

## 2. Materials and Methods

### 2.1. Patient Population

We conducted a retrospective study including patients diagnosed with bladder cancer between January 2005 and December 2020 in a single tertiary medical center. The medical charts of these patients were reviewed. Only those diagnosed with NMIBC administered intravesical BCG instillation therapy after TUR-BT were enrolled. Patients without contraindications such as bleeding with clots or bladder perforation received immediate single instillation (SI) of intravesical chemotherapy within 24 h after the TUR-BT. We classified these patients based on EAU NMIBC prognostic factor risk groups [8]. The low-risk group was excluded since an SI was considered treatment complete in this group, and EAU guidelines suggested no further intravesical instillation was needed. Patients with previous urothelial carcinoma history (including bladder or upper urinary tract) or other malignancy history were eligible. This study was approved by the Institutional Review Board (IRB) of Chang Gung Memorial Hospital. The IRB of Chang Gung Memorial Hospital waived the requirement for patient consent to review their medical records due to the study’s retrospective nature. The data collection confidentiality agreement fulfilled the requirements of the Declaration of Helsinki.

### 2.2. Risk Group Classification

Based on the EAU guideline and the 2004/2016 WHO histological grading system [8], we defined low risk as a primary, solitary, Ta, papillary urothelial neoplasm of low malignant potential (PUNLMP) or low grade (LG) tumor with a size smaller than 3 cm and no carcinoma in situ (CIS). On the other hand, the high-risk group was defined as any of the following: T1, high grade (HG), or CIS. Ta LG tumors with multiple, recurrent, and large (>3 cm) features all presented were regarded as high-risk. Finally, tumors that could not be classified into the low or high-risk category were defined as intermediate risk.

### 2.3. Data Collection

Patients’ general characteristics were collected: gender, age, body height, body weight, body mass index (BMI), smoking habit, comorbidity, and cancer history. We also identified tumor-related parameters, including tumor stage, histology, pathological grade, tumor size, tumor number, and risk group. Intravesical instillation data were also recorded, including the number of BCG, the alternative chemotherapy instillation if it existed, and the number of total intravesical instillations (IVI).

### 2.4. Endpoints

Local recurrence of bladder cancer was set as the primary endpoint, while patients were followed for recurrence-free survival until the final analysis. Recurrence-free survival (RFS) was defined as the period from diagnosis to the time of bladder recurrence. In the absence of confirmation of bladder recurrence, RFS time was censored at the last date at which the patient was known to be recurrence-free. Secondary endpoints were progression to muscle-invasive disease, progression-free survival, the event of death, and overall survival. Moreover, the reason for BCG discontinuation and the appearance of side effects, including local or systemic, were identified.

### 2.5. Follow Up

All patients received regular follow-ups with cystoscopy every three months in the first two years, then every six months for another two years, and annually.

### 2.6. Intravesical Instillation Dosage and Protocol

The protocol of BCG instillation consisted of six-weekly instillation as an induction course and one-year maintenance course (three weekly instillations at months 3, 6, and 12) for the intermediate-risk group. As for the high-risk group, the maintenance course would extend to three years, with an additional three weekly instillations at months 18, 24, 30, and 36. The dose of BCG was 81 mg during each instillation. The BCG strains were BCG-TICE (OncoTICE^®^, MSD, Rahway, NJ, USA). The chemotherapy agents used in the alternative intravesical instillation were epirubicin (50 mg of each dose) and mitomycin C (40 mg of each dose). Only a single agent was used during each instillation.

### 2.7. Statistical Analysis

We analyzed the correlation of each parameter with Pearson Chi-square, Fisher’s exact test, and the independent-T test. Recurrence-free, progression-free, and overall survival were analyzed with the Kaplan–Meier survival test. We regarded *p*-values less than 0.05 as significant. Univariate and multivariate analyses were performed using log-rank tests and Cox proportional hazard regression. Significant variables (*p* < 0.05) or neared significance (*p* < 0.2) in univariate analysis were included in the multivariate analysis. All statistical analyses were performed using IBM SPSS Statistics 26.00 software (IBM Corp., Armonk, NY, USA).

## 3. Results

A total of 3351 patients with bladder cancer were initially assessed. After excluding MIBC and those without BCG instillation records, 497 patients with NMIBC and receiving BCG intravesical instillations were enrolled, with only 17 patients completing the whole course of BCG instillation. Among the 480 patients who had BCG instillation interruption, 421 completed the induction course but dropped out during the maintenance course, and others failed to complete the induction course. A total of 90 patients received alternate chemotherapy instillation after BCG discontinuation. The enrollment flowchart is illustrated in Figure 1.

Among the enrolled patients, the male-to-female ratio was 3.40, and the mean age at diagnosis was 66.7 years. Ninety-two patients (18.5%) had a history of bladder cancer. Of all included patients, 40.4% and 53.9% had stage Ta and T1, respectively, while only 5.6% had pure Tis stage. Histological high-grade tumors accounted for 75.9%. Four hundred thirteen patients (83.1%) were categorized in the high-risk group. The mean follow-up period was 70.8 ± 46.0 months. The detailed characteristics of the patients, such as BMI, comorbidity and cancer history, and other tumor parameters, are listed in Table 1.

Comparison analysis was performed between the groups using intravesical chemotherapy as an alternative after BCG discontinuation (short as [BCG + Chemo]) and the pure BCG group (short as [BCG only]). The general characteristics and tumor-related parameters were analyzed, with no statistical difference in most parameters except in instillation counts (Table 2). The [BCG only] patients received more BCG instillation than [BCG + Chemo] group (10.5 vs. 7.3, *p* < 0.001), but the latter had statistically more total IVI counts (10.5 vs. 14.9, *p* < 0.001).

During a median follow-up duration of 59.3 months (range: 2.6–227.1 months), the all-time bladder recurrence-free survival (RFS) revealed a better trend in the [BCG + Chemo] group than the [BCG only] group. However, it did not reach the statistical difference (log-rank test: *p* = 0.072). As we focused on the three-year RFS, the [BCG + Chemo] group showed a significantly higher survival rate than the [BCG only] group (log-rank test: *p* = 0.022). The progression-free survival (PFS) and overall survival (OS) between these two groups were analyzed, and no statistical difference was found (log-rank test: *p* = 0.275 and 0.075, respectively). The Kaplan–Meier survival plots are illustrated in Figure 2.

Univariate and multivariate analyses using the Cox regression method were performed to evaluate the effect of various parameters on recurrence events. Both all-time recurrence and three-year recurrence share similar independent factors: diabetes mellitus, chronic kidney disease, history of bladder cancer, tumor stage and number, risk group category, and instillation methods. The [BCG only] method was significantly associated with increased all-time recurrence (HR = 2.015; 95% CI 1.169–3.474) and three-year recurrence (HR = 2.148; 95% CI 1.208–3.819) relative to the [BCG + Chemo] method (Table 3).

Among the population of BCG interruption, BCG shortage accounted for 21.3% of the reasons for discontinuation, while 21.0% were due to the side effects. About 42.8% of the stoppage was due to individual requirements without objective toxicity events. Side effects could be further divided into systemic (30.7%) and local effects (69.3%), with the former including fever and malaise, and the latter consisting of lower urinary tract symptoms (LUTS) and urinary tract infection (UTI). The details are illustrated in Figure 3.

## 4. Discussion

The current consensus on the maintenance protocol of BCG instillations was based on several randomized trials and meta-analyses [9,10,11,12,13], which is superior to BCG induction alone or chemotherapy instillation in reducing recurrences of bladder cancer. BCG toxicity has been well recognized during the long instillation protocol and has gained more attention. In the BCG toxicity report of an EORTC randomized study [14], the proportion of local and systemic side effects is about 2:1, similar to our study results. Further trials were conducted to achieve less toxicity, without affecting patient prognosis [15].

The EORTC 30962 trial was designed with randomization of its study population by dosage (one-third dose or full dose) and duration of maintenance (one or three years) [16]. The results showed no statistical differences in toxicity among these treatment groups. Neither reducing the dose nor shortening the maintenance duration would lead to less toxicity. The NIMBUS trial showed a reduced frequency of BCG schedule of nine instillations within twelve months, which was inferior to the standard protocol (15 instillations) in recurrence-free intervals, and therefore being early halted by the EAU Research Foundation [6]. Another modified BCG maintenance regimen, as one BCG instillation every three months following induction therapy, was evaluated in the CUETO 98013 trial, which showed no benefit in reducing recurrence and progression rates than induction therapy alone [17].

In addition to a marked side effect of BCG, another reason for treatment stoppage is BCG shortage. Between 2013 and 2016, the closure of BCG production factories led to a worldwide shortage of BCG. A retrospective study evaluated the impact of the shortage by comparing the bladder cancer recurrence rate before and after the shortage. A higher recurrence rate was found in the group with instillation treatment during the BCG shortage period [18]. Due to the COVID-19 epidemic, the role of BCG as a nonspecific immune stimulant was re-emphasized, increasing the global demand. This growing use and need for BCG led to another episode of worldwide shortage and was declared by American Urological Association (AUA) announcement [19].

Besides changing the regimen of BCG instillation, using chemotherapy as an intravesical instillation agent was acceptable in NMIBC treatment. Adjuvant chemotherapy instillation significantly reduced the recurrence rate than TUR-BT alone [20]. Comparison between BCG and chemotherapy instillation has been well studied, and chemotherapy appeared inferior to BCG maintenance therapy in terms of recurrence prevention with fewer side effects [13,21,22]. Some studies have reported device-assisted intravesical chemotherapy such as microwave-induced chemohyperthermia or electromotive methods to amplify the cytotoxic effects of chemotherapy on bladder tumor cells, and the results were promising. One RCT using MMC with microwave-induced hyperthermia method presented a significantly higher 24-month RFS than BCG in the intermediate- and high-risk papillary NMIBC [23]. However, these methods encountered challenges in administration due to the need for expensive machines and trained employees [23,24,25,26].

This study has focused on the population who received initial BCG instillation but faced stoppage either due to BCG shortage or intolerance to toxicity. Shifting to intravesical chemotherapy instillation during BCG discontinuation seemed feasible, and our results support this hypothesis. We found that chemotherapy as an alternative would have significantly higher three-year recurrence-free survival than stopping further instillation. Moreover, shifting to chemotherapy instillation is an independent protecting factor for three-year and all-time recurrence. Most importantly, this strategy does not need an extra device and could reduce recurrence without increasing side effects. This easily applicable feature makes our findings worth more emphasis.

One limitation of this study was the retrospective study design, which might affect the evidence level. Despite equivalent results between subgroups analysis, selection bias should also be considered as there might be implicit factors that make people more willing to receive different instillation agents than stop treatment. Another limitation is the small target population size, with only 90 patients receiving BCG and alternative chemotherapy instillation. This might be due to the multiple selections during inclusion, as the cohort should have BCG instillation first with an incomplete treatment course, then have chemotherapy instillation as an alternative. Finally, issues such as comparing the [BCG + Chemo] group versus the BCG protocol complete group or cancer-specific survival between different instillation strategies remain unclear. Further sample size expansion or prospective study would lead to a more precise comprehension of these issues.

## 5. Conclusions

For patients with intermediate- and high-risk NMIBC who received BCG instillation and have faced treatment stoppage, subsequent replacement with chemotherapy agents has a better three-year RFS. Our strategy could still be practiced simply and efficiently in reducing the bladder recurrence rate.

## Figures and Tables

**Figure 1 cancers-15-01345-f001:**
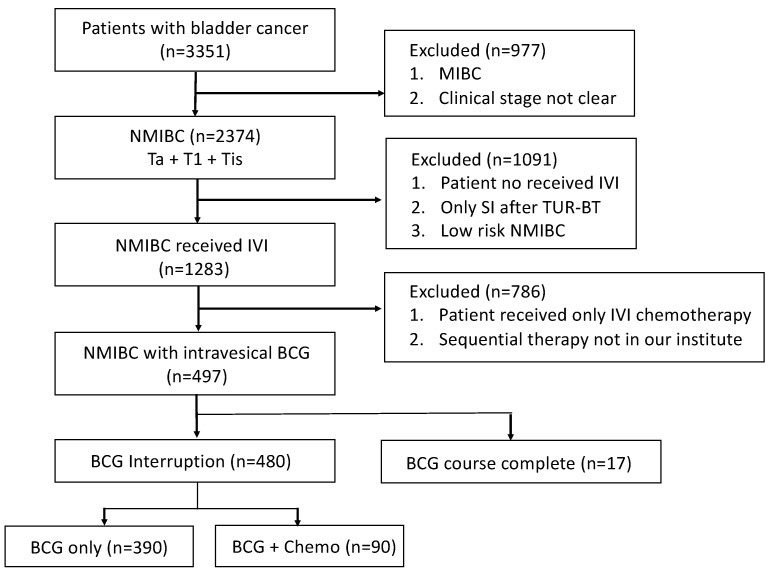
Flowchart of the patient enrollment process of the study cohort. Patients with NMIBC and BCG intravesical instillation record. Those with incomplete instillation were identified and divided into different subgroups based on their instillation regimen.

**Figure 2 cancers-15-01345-f002:**
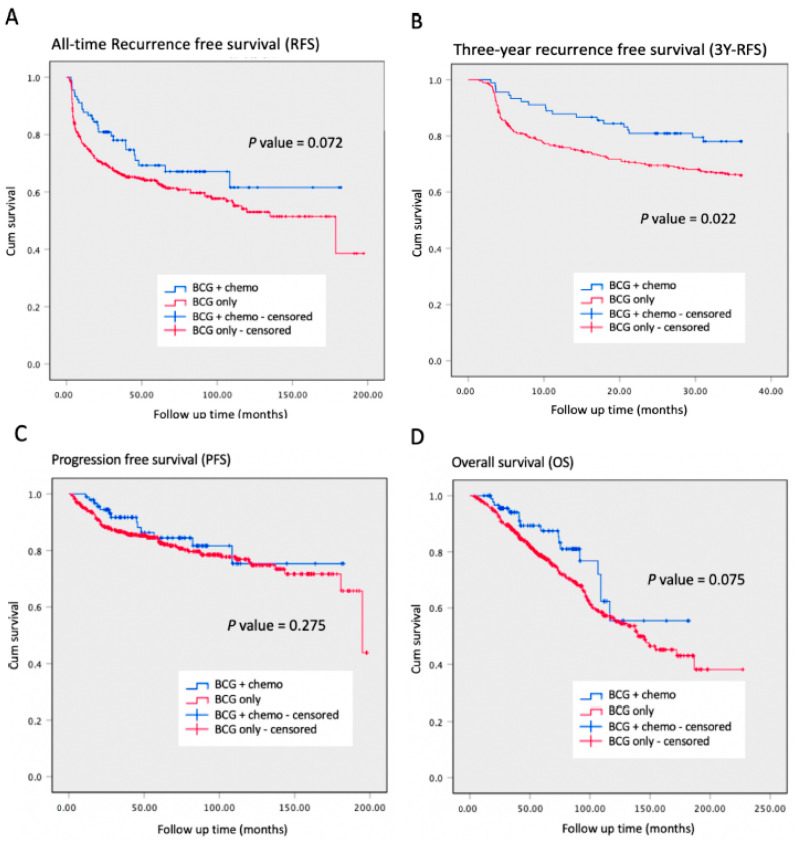
Kaplan–Meier survival curve with end point analysis between instillation subgroups. The survival analysis between [BCG + Chemo] and [BCG only] displaying (**A**) time to recurrence during the whole follow-up period, (**B**) time to recurrence within three years, (**C**) time to progression during the whole follow-up period, and (**D**) time to death during the whole follow-up period.

**Figure 3 cancers-15-01345-f003:**
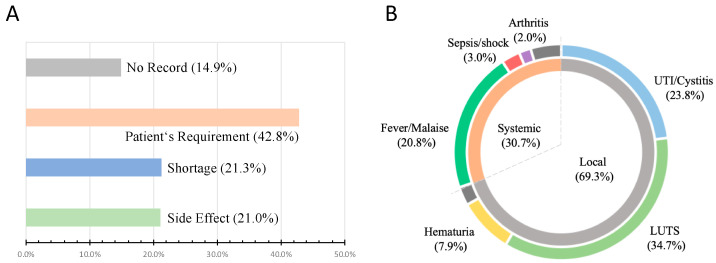
Causes of BCG discontinuation and details of side effects. The proportion of reasons for BCG discontinuation presented in (**A**), (**B**) describes the details of side effects, including classification and proportion of significant symptoms of BCG toxicity.

**Table 1 cancers-15-01345-t001:** Patients’ general characteristics and tumor-related parameters.

Variable	Mean/Number	SD	Range/Percentage
Patients’ general characteristics			
Total number	497		
Sex			
Male	384		77.3%
Female	113		22.7%
Age (years)	66.7	11.2	21–94
Height (cm)	161.7	8.3	137–182
Weight (kg)	64.9	12.2	27–130
BMI (kg/cm^2^)	24.7	3.8	12.8–50.8
Smoke	185		37.2%
Performance Status			
ECOG 0	355		71.4%
ECOG 1	127		25.6%
ECOG 2	15		3%
Comorbidity			
CVA/TIA	32		6.4%
CAD	47		9.5%
Hypertension	270		54.3%
Diabetes Mellitus	122		24.5%
CKD (moderate-severe)	83		16.7%
Cirrhosis	15		3%
History of Cancer			
Bladder Cancer	92		18.5%
UTUC	61		12.3%
Other Cancer	45		9.1%
Tumor-related parameters			
T stage			
Ta	201		40.4%
T1	268		53.9%
Tis	28		5.6%
Concomitant Cis			
No	463		93.2%
Yes	34		6.8%
Grade			
Low	98		19.7%
High	377		75.9%
not applicable	22		4.4%
Tumor number			
Single	190		38.2%
Multiple	307		61.8%
Tumor size			
≤3 cm	283		56.9%
>3 cm	75		15.1%
no record	139		28.0%
Risk group			
Intermediate	84		16.9%
High	413		83.1%
Interruption			
During induction	59		11.9%
During maintenance	421		84.7%
No	17		3.4%
Instillation agent			
BCG only	407		81.9%
BCG + chemo	90		18.1%
Follow-up (months)	70.8	46.0	

BMI = Body Mass Index; ECOG = Eastern Cooperative Oncology Group Performance Status; CVA/TIA = Cerebrovascular Accident/Transient Ischemic Attack; CAD = Coronary Artery Disease; CKD = Chronic Kidney Disease; UTUC = Upper Tract Urothelial Carcinoma; Cis = Carcinoma in situ; BCG = Bacillus Calmette-Guérin; SD = Standard Deviation.

**Table 2 cancers-15-01345-t002:** Difference analysis between instillation subgroups.

	BCG + Chemo (N = 90)		BCG only (N = 390)		*p* Value
	Mean/Number	SD/Percentage	Mean/Number	SD/Percentage	
Gender					
Male	65	72.2%	308	79.0%	0.165
Female	25	27.8%	82	21.0%	
Age (years)	65.5	9.4	67.0	11.6	0.156
Body height (cm)	162.1	8.7	161.7	8.0	0.657
Body weight (kg)	66.0	14.3	64.7	11.7	0.365
BMI (kg/cm^2^)	25.0	4.7	24.6	3.6	0.412
Smoke	30	33.3%	148	37.9%	0.414
Performance Status					
ECOG 0	61	67.8%	280	71.8%	0.449
ECOG ≥ 1	29	32.2%	110	28.2%	
Comorbidity					
CVA/TIA	6	6.7%	25	6.4%	0.929
CAD	7	7.8%	39	10.0%	0.519
Hypertension	54	60.0%	210	53.8%	0.29
Diabetes Mellitus	25	27.8%	94	24.1%	0.467
CKD (moderate-severe)	18	20.0%	65	16.7%	0.451
Cirrhosis	3	3.3%	11	2.8%	0.733
Cancer history					
Bladder Cancer	17	18.9%	71	18.2%	0.88
UTUC	8	8.9%	51	13.1%	0.275
Other Cancer	12	13.3%	32	8.2%	0.129
T stage					
Ta	29	32.2%	156	40.0%	0.363
T1	56	62.2%	211	54.1%	
Tis	5	5.6%	23	5.9%	
Concomitant Cis					
No	81	90.0%	365	93.6%	0.231
Yes	9	10.0%	25	6.4%	
Grade					
Low	13	14.4%	68	17.4%	0.62
High	74	82.2%	303	77.7%	
not applicable	3	3.3%	19	4.9%	
Tumor number					
Single	31	34.4%	157	40.3%	0.309
Multiple	59	65.6%	233	59.7%	
Tumor size					
≤3 cm	52	57.8%	222	56.9%	0.988
>3 cm	13	14.4%	58	14.9%	
no record	25	27.8%	110	28.2%	
Risk group					
Intermediate	10	11.1%	57	14.6%	0.387
High	80	88.9%	333	85.4%	
Instillation Agents					
BCG	7.3	4.1	10.5	4.8	<0.001
Chemo	7.6	5.2	-		<0.001
Total IVI	14.9	5.2	10.5	4.8	<0.001
Follow-up (months)	64.5	39.5	71.2	47.0	0.215

BMI = Body Mass Index; ECOG = Eastern Cooperative Oncology Group Performance Status; CVA/TIA = Cerebrovascular Accident/Transient Ischemic Attack; CAD = Coronary Artery Disease; CKD = Chronic Kidney Disease; UTUC = Upper Tract Urothelial Carcinoma; IVI = intravesical instillation; Cis = Carcinoma in situ; BCG = Bacillus Calmette-Guérin; SD = Standard Deviation.

**Table 3 cancers-15-01345-t003:** Univariate and multivariate Cox regression analysis of all-time recurrence and three-year recurrence.

.	All-Time Recurrence				Three-Year Recurrence			
	Univariate		Multivariate		Univariate		Multivariate	
	Hazard Ratio (95% CI)	*p* Value	Hazard Ratio (95% CI)	*p* Value	Hazard Ratio (95% CI)	*p* Value	Hazard Ratio (95% CI)	*p* Value
Gender								
Male	reference		-		reference		-	
Female	0.822 (0.546–1.239)	0.350			0.801 (0.507–1.264)	0.340		
Age (years)	1.002 (0.986–1.017)	0.835	-		1.006 (0.989–1.023)	0.508	-	
BMI (kg/cm^2^)	0.949 (0.908–0.992)	0.021	-		0.962 (0.916–1.01)	0.116	-	
Smoke	1.159 (0.82–1.639)	0.403	-		1.213 (0.83–1.771)	0.318	-	
Performance Status								
ECOG 0	reference		-		reference		-	
ECOG ≥ 1	1.087 (0.77–1.533)	0.636			0.99 (0.678–1.446)	0.957		
Comorbidity								
CVA/TIA	1.65 (0.942–2.891)	0.080	-		1.369 (0.716–2.619)	0.343	-	
CAD	0.808 (0.49–1.332)	0.403	-		0.749 (0.422–1.329)	0.323	-	
Hypertension	1.22 (0.868–1.715)	0.253	-		1.1 (0.76–1.592)	0.614	-	
Diabetes Mellitus	1.361 (0.953–1.943)	0.090	1.592 (1.009–2.511)	0.046	1.401 (0.955–2.056)	0.085	1.732 (1.092–2.746)	0.020
CKD (moderate-severe)	1.793 (1.251–2.569)	0.001	3.116 (1.838–5.281)	<0.001	1.493 (0.99–2.251)	0.056	1.899 (1.121–3.217)	0.017
Cirrhosis	0.379 (0.118–1.212)	0.102	-		0.46 (0.143–1.477)	0.192	-	
Cancer history								
Bladder Cancer	2.428 (1.697–3.473)	<0.001	2.335 (1.408–3.873)	0.001	2.408 (1.639–3.536)	<0.001	2.565 (1.549–4.249)	< 0.001
UTUC	1.271 (0.811–1.992)	0.295	-		1.228 (0.756–1.996)	0.407	-	
Other Cancer	1.05 (0.614–1.796)	0.858	-		1.19 (0.691–2.047)	0.530	-	
T stage								
Ta	reference		reference		reference		reference	
T1/Tis	2.377 (1.582–3.572)	<0.001	2.895 (1.727–4.854)	<0.001	2.357 (1.498–3.707)	<0.001	2.634 (1.547–4.486)	<0.001
Grade								
Low	reference		-		reference		-	
High	2.468 (0.749–8.137)	0.138			1.789 (0.543–5.896)	0.339		
Concomitant Cis								
No	reference		reference		reference		-	
Yes	2.604 (1.280–5.297)	0.008	3.001 (1.413–6.370)	0.004	1.441 (0.701–2.963)	0.320		
Tumor number								
Single	reference		reference		reference		reference	
Multiple	1.573 (1.134–2.183)	0.007	1.679 (1.098–2.568)	0.017	1.514 (1.055–2.174)	0.024	1.650 (1.064–2.557)	0.025
Tumor size								
≤3 cm	reference		-		reference		-	
>3 cm	0.799 (0.484–1.318)	0.379			0.835 (0.496–1.405)	0.497		
Risk group								
Intermediate	reference		reference		reference		reference	
High	0.799 (0.484–1.318)	0.379	2.992 (1.495–5.990)	0.002	4.371 (1.196–15.976)	0.026	2.814 (1.396–5.671)	0.004
Instillation Methods								
BCG + Chemo	reference		reference		reference		reference	
BCG only	1.572 (0.954–2.590)	0.076	2.015 (1.169–3.474)	0.012	1.826 (1.055–3.160)	0.032	2.148 (1.208–3.819)	0.009

BMI = Body Mass Index; ECOG = Eastern Cooperative Oncology Group; CVA/TIA = Cerebrovascular Accident/Transient Ischemic Attack; CAD = coronary artery disease; CKD = Chronic Kidney Disease; Cis = Carcinoma in situ; BCG = Bacillus Calmette-Guérin; UTUC = Upper Tract Urothelial Carcinoma; CI = Confidence Interval.

## Data Availability

Data available on request due to restrictions e.g., privacy or ethical. The data presented in this study are available on request from the corresponding author. The data are not publicly available due to their containing information that could compromise the privacy of research participants.

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
