# Peer review of "Replacement Instead of Discontinuation of Bacillus Calmette-Guérin Instillation in Non-Muscle-Invasive Bladder Cancer"

_cancers, 2023, doi:10.3390/cancers15041345_

Round 1
Reviewer 1 Report
This is a well-written and interesting paper with novel and significant results that should impact daily clinical practice.
A few minor questions occur and should be addressed:
1. Why did all patients receive immediate/early instillation therapy? This is not recommended by current guidelines. Is this common practice in your institution? Please explain.
2. Why were there only 90 patients who received BCG + chemo? This should be addressed and added in the discussion section.
3. Do you have data on concomitant Cis? I would include this as "yes" or "no" in Table 1 and the univariate analysis.
4. Do you have data on WHO1973 grading?
5. There is a difference in mean follow-up between text (70.8, p.4) and table 1 (59.3, p. 5). Please check again and correct.
Reviewer 2 Report
First of all, I would like to complement the authors for their paper. They describe a significant better RFS in a chemotherapy replacement group, than no instillation at all, after incomplete BCG instillations. It is a pity, that no comparison is made between the BCG+chemotherapy group and the BCG-completion group. this would be really interesting.
a few comments to clearify some aspects of these data:
- this is a retrospective desigen from 2005-2020. the authors state that BCG stoppage is mainly due to shortages. however shortages of BCG started to occur in 2012-2013....what about the 2005-2012 group. how many drop-outs did occur and what was the reason.
-In the paper it remains unclear how many pts did complete the induction course and how many dropped out during maintenance course.
- in 2.1 the authors state that all pts received one cycle of chemotherapy post-op...all?? not one had heamaturia? perforation?. in Fig 1. the autohors state that there were exluded pts (1091) of some pt without IVI.
- what was the definition of "local recurrence" (alinea 2.4). If a pt ha HR NMIBC and after induction BCG no continuation of IVI, than made an pTaGI recurrence? whas this counted as a recurrence? if so, than a Will-Roger phenomenon could occur if these (not-significant) recurences were left out of analysis, and could improve the outcomes of the BCG-discontinuation group, as well for the BCG+chemotherapy group.
- 3. results, r 136. "....17 pts receiving BCG instillation." missing the word: full?
- in fig 1 BCG cours completion (3yrs?) is ony 17 pts, this is 3% of the pts who started BCG. only 21.3% stopped due to shortages of BCG. so around 80% stopped due to side-effects? this is not what I see in my practice, or what literature suggests...why so low? whas travel-distance a problem, reimbursement issues? were there costs (travel/therapy) for the patients?
- did no one tried one-third dosis of BCG? we applied, in times of shortages, 1/3 dose to all pts, just to help more pts and divide the BCG among them. this was also recommended by the EAU. Is this option considered? did pts received 1/3 dose BCG? due to shortages or to side-effects?
R195-196"...important reasons include individual requirements without toxicity events"...What does this mean???
- fig 3B shows us that 24% stopped due to a urinary tract infection. however, we dispone a week, give antibiotics en continue with BCG. was this strategy applied in your centre? or was everybody with a UTI advised to stop BCG?
r 240-241: it is true that chemohyperthermia (CHT) and/or electromotive methods are more challenging, not due to the schedule (these are similar to MMC schedule), but due to the need of an expensive machine and trained employees.
furthermore, ref. 23-26 in general, do! show promising results (r 240). even in RCT-setting (i.e. Arends et al 2016 and Di Stasi et al 2006). Arends et al., for example, shows that CHT is a safe and effective treatment option in patients with intermediate- and high-risk papillary NMIBC. A significantly higher 24-mo RFS in the CHT group was seen. Based on these results CHT is an option for BCG therapy as adjuvant treatment for intermediate- and high-risk papillary NMIBC. ALSO IN TIMES OF BCG SHORTAGES. however, as previously mentioned, you do need trained employees and an expensive machine.
